# Genomic Features and Molecular Function of a Novel Stress-Tolerant *Bacillus halotolerans* Strain Isolated from an Extreme Environment

**DOI:** 10.3390/biology10101030

**Published:** 2021-10-12

**Authors:** Xiaohui Wu, Huijun Wu, Ruoyi Wang, Zhengqi Wang, Yaming Zhang, Qin Gu, Ayaz Farzand, Xue Yang, Mikhail Semenov, Rainer Borriss, Yongli Xie, Xuewen Gao

**Affiliations:** 1Key Laboratory of Integrated Management of Crop Diseases and Pests, Ministry of Education, Department of Plant Pathology, College of Plant Protection, Nanjing Agricultural University, Nanjing 210095, China; 2019202006@njau.edu.cn (X.W.); hjwu@njau.edu.cn (H.W.); 2019102046@njau.edu.cn (R.W.); 2019102045@njau.edu.cn (Z.W.); 2019802211@njau.edu.cn (Y.Z.); guqin@njau.edu.cn (Q.G.); dr.ayaz21@outlook.com (A.F.); 2Department of Grassland Science, College of Agricultural and Husbandry, Qinghai University, Xining 810016, China; skryangxue@163.com; 3State Key Laboratory of Plateau Ecology and Agriculture, Qinghai University, Xining 810016, China; 4Department of Soil Biology and Biochemistry, Dokuchaev Soil Science Institute, 119017 Moscow, Russia; mikhail.v.semenov@gmail.com; 5Institut für Biologie, Humboldt Universität Berlin, 10115 Berlin, Germany; 6Nord Reet UG, Marienstr. 27a, 17489 Greifswald, Germany

**Keywords:** Qinghai–Tibet Plateau, *Bacillus halotolerans*, stress tolerance, biocontrol, genomic features, comparative genomic analysis

## Abstract

**Simple Summary:**

The Qinghai–Tibet Plateau is known as the “third pole of the world”. Due to the extreme geographical location, Qinghai–Tibet Plateau has unique ecosystems characterized by oxygen deficiency, low temperature, high salinity and alkalinity. We carried out the current study to explore the excellent extremophilic *Bacillus* strains via potential stress resistance as well as biocontrol properties in the Qinghai–Tibet Plateau. We found a *Bacillus halotolerans* strain with a promising ability to withstand harsh environments and which also exhibits an optimistic biocontrol activity against plant pathogens. We revealed the whole genome sequencing and its taxonomic position and elucidated its molecular functions that were responsible for enhancing stress tolerance as well as suppressing plant pathogens at the genetic level. Lastly, we identified this strain harbored the specific genes associated with stresses resistance, biocontrol function, and can be used as a biological agent in the agriculture field.

**Abstract:**

Due to its topographical position and climatic conditions, the Qinghai–Tibet Plateau possesses abundant microorganism resources. The extremophilic strain KKD1 isolated from Hoh Xil possesses strong stress tolerance, enabling it to propagate under high salinity (13%) and alkalinity (pH 10.0) conditions. In addition, KKD1 exhibits promising biocontrol activity against plant pathogens. To further explore these traits at the genomic level, we performed whole-genome sequencing and analysis. The taxonomic identification according to the average nucleotide identity based on BLAST revealed that KKD1 belongs to *Bacillus halotolerans*. Genetic screening of KKD1 revealed that its stress resistance mechanism depends on osmotic equilibrium, membrane transportation, and the regulation of ion balance under salt and alkaline stress. The expression of genes involved in these pathways was analyzed using real-time quantitative PCR. The presence of different gene clusters encoding antimicrobial secondary metabolites indicated the various pathways by which KKD1 suppresses phytopathogenic growth. Moreover, the lipopeptides surfactin and fengycin were identified as being significant antifungal components of KKD1. Through comparative genomics analysis, we noticed that KKD1 harbored specific genes involved in stress resistance and biocontrol, thus providing a new perspective on the genomic features of the extremophilic *Bacillus* species.

## 1. Introduction

The Qinghai–Tibet Plateau (QTP), the largest plateau on earth, with the highest average altitude, is known as the “third pole of the world” [1]. Due to its extreme geographical location and climatic conditions, the QTP has unique extreme ecosystems, characterized by oxygen deficiency, low temperatures, and intense ultraviolet radiation [2]. The rhizosphere of plants growing in the extreme environment could be a rich source of microbes with potential stress resistance and biocontrol properties [3]. *Bacillus* and *Pseudomonas* are potential biological control agents [4], and the application of biopesticide is an environmentally friendly alternative to chemical fertilizers [5].

Numerous studies have demonstrated that most biocontrol strains cannot propagate under stressful conditions [6]. A low survival rate is a major challenge for the application of microbial products in extreme environments. The ability to withstand harsh environments is thus critical to microbial survival. Extremophilic strains are continuously confronted with the physicochemical changes associated with extreme environments [3]. We previously isolated numerous *Bacillus* strains from the QTP. Some of them displayed excellent stress tolerance. The *Bacillus* sp. RJGP41 and CJCL2 isolated from Tibet have the ability to propagate at 4 °C, while *B. safensis* GBSW22 and *B. zhangzhouensis* LLTC93 are able to grow between 10 °C and 50 °C [3]. In our previous work, we only reported on the strains that possessed cold and heat stress resistance. Little is known about the tolerance of these strains to salt and alkali stress, including their associated defense mechanisms.

Phytopathogenic fungi severely reduce the yield of agricultural products, and the use of biological fertilizer offers a sustainable way of avoiding this problem [7]. The ability to produce diverse secondary metabolic compounds is the most outstanding biocontrol feature of the *Bacillus* species. Approximately 10% of the genes in the genome of the model biocontrol strain *B. velezensis* FZB42 are involved in the production of secondary metabolites, including polyketides, antimicrobial peptides and lipopeptides [8]. Lipopeptides are synthesized by non-ribosomal peptide synthetases (NRPS) that are composed of large multi-enzyme complexes, and which play an important role in inhibiting phytopathogenic pathogens [9]. Preliminary results suggest that stress-tolerant representatives of the genus *Bacillus* play a major role in QTP ecosystems [6], and their biocontrol capability has been reported by our laboratory. The *Bacillus* strains isolated from Tibet such as *B. halotolerans* DGL6 and *B. velezensis* GBSW11 and DJFZ40 showed strong antagonistic activities towards *Phytophthora capsica*, *Xanthomonas oryzae,* and *Sclerotinia sclerotiorum* [10].

The objective of this study was to isolate stress-tolerant *Bacillus* spp. from different plant rhizospheres in the QTP and analyze their stress resistance and biocontrol potential at the molecular level. We report here the complete genome of the novel biocontrol strain *B. halotolerans* KKD1, which was found to be closely related to the mesophilic *B. subtilis* type strain, but is distinguished by its ability to withstand harsh environmental conditions. Moreover, we revealed its taxonomic position and found that its molecular features were responsible for its capacity to enhance stress tolerance and to suppress plant pathogens. At last, we noticed that KKD1 harbored specific genes associated with stress resistance and biocontrol functions, and that it could be used as a perspective biological agent in agriculture.

## 2. Materials and Methods

### 2.1. Collection of Samples and Isolation of Bacteria

The samples were collected from the rhizosphere of different plants (*Brassica napus, Tamarix ramosissima, Nitraria tangutorum, Kobresia pygmaea, Phragmites communis, Peganum harmala, Androsace umbellata, Suaeda glauca*) growing in extreme habitats in the QTP (Appendix A). The samples were stored at 4 °C before further processing. Ten grams of the soil samples was added to 90 mL of saline solution and incubated with shaking at 200 rpm and 37 °C for 30 min. After centrifugation, the sediment was diluted (10^−1^, 10^−2^, and 10^−3^) and then incubated for 10 min at 80 °C to enrich spore-forming bacteria, following which it was streaked onto LB plates at 37 °C for 12 h. Single colonies were re-streaked several times in order to avoid contaminated clones [3].

### 2.2. DNA Isolation and gyrB Gene Sequence Analysis

Total genomic DNA was extracted from bacterial cells using the DNA Extraction kit D3350-01 (Omega Bio-Tek, Norcross, GA, USA) according to the manufacturer’s instructions. The purity and quantity of the extracted DNA were evaluated using the NanoDrop spectrophotometer 1000 (Thermo Fisher Scientific, 81 Wyman Street, Waltham, MA, USA). The *gyr**B* gene sequence was amplified with an appropriate primer pair (Appendix A). The amplification profile was: 95 °C for 3 min, 32 cycles of 95 °C for 15 s, 56 °C for 30 s, 72 °C for 1 min, and the final extension at 72 °C for 3 min. The *gyrB* sequences were determined by the Sanger methodology. Bacteria with closely related sequences were identified using the Advanced BLAST search program of NCBI [3]. The neighbor-joining (NJ) tree was constructed from the *gyrB* sequences using MEGA 7.0 with the p-distance model (https://mega.software.informer.com/; accessed on 30 June 2021) [11]. NCBI accession numbers are listed in Appendix A.

### 2.3. Genome Sequencing and Analysis

Total genomic DNA was obtained as described in Section 2.2. The whole genomes of strains KKD1 (GenBank: CP054584.1, Ref seq: NZ_CP054584.1) and KKLW (GenBank: CP054714.1, Ref seq: NZ_CP054714.1) were sequenced on a PacBio RSII platform (BGI Genomics Co., Ltd.). The genome coverage was 510× (CP054584.1). De novo assembly was performed using Canu v. 1.3 software to obtain the complete circular genome. Automatic gene annotation was performed using the NCBI Prokaryotic Genome Annotation Pipeline (PGAP) [11].

### 2.4. Comparative Genome Analysis

The CRISPR-Cas system, transfer RNA (tRNA) genes, and ribosome RNA (rRNA) genes were predicted by the MinCED (https://github.com/ctSkennerton/minced/tree/master; accessed on 25 July 2020), tRNAscan-SE (http://lowelab.ucsc.edu/tRNAscan-SE/; accessed on 25 July 2020), and Barrnap (https://github.com/tseemann/barrnap; accessed on 25 July 2020), respectively. Gene islands were analyzed by IslandPath-DIMOB (http://pathogenomics.sfu.ca/islandviewer; accessed on 25 July 2020). Gene prediction was carried out by GeneMarks (http://topaz.gatech.edu/GeneMark/; accessed on 25 July 2020), and Prodigal (https://github.com/hyattpd/Prodiga; accessed on 25 July 2020). The annotation of gene functions and metabolic systems was based on the Carbohydrate-Active enZYmes (CAZy) database, (http://www.cazy.org/; accessed on 25 July 2020), Cluster of Orthologous Groups of Proteins (COG, http://www.ncbi.nlm.nih.gov/COG; accessed on 26 August 2020), Kyoto Encyclopedia of Genes and Genomes (KEGG, https://www.kegg.jp/; accessed on 26 August 2020), Gene Ontology (GO, http://geneontology.org/; accessed on 26 August 2020), Swiss-Prot (http://www.ebi.ac.uk/swissprot/; accessed on 26 August 2020), and Non-Redundant Protein Database (Nr, https://www.ncbi.nlm.nih.gov/; accessed on 26 August 2020). Gene clusters involved in the synthesis of secondary metabolites were identified using the online tool antiSMASH 5.0 (https://antismash.secondarymetabolites.org/; accessed on 28 September 2021). Phylogenetic trees were constructed using the neighbor-joining method in MEGA v7.0 software and the iTOL v5 online tool (https://itol.embl.de/; accessed on 15 June 2021). The target genes of the sRNAs were predicted by the online tool TargetRNA2 (http://cs.wellesley.edu/~btjaden/TargetRNA2/; accessed on 26 August 2020). Mummer software was used for the detection and evolutionary analysis of gene synteny and collinearity.

### 2.5. Inhibition of Phytopathogenic Fungi

In brief, a paper disk (7 mm in diameter) containing the freshly grown fungal pathogen (*Fusarium graminearum*) was placed into the center of a Potato Dextrose Agar (PDA) plate. The plates were incubated at 25 °C for 2 d. Then, 5 μL of exponentially growing bacterial cells was inoculated on sterile filter paper discs (4 mm), which were separately placed in the plates in a cruciform arrangement. The plates were incubated for 3 d at 25 °C. The diameters of the inhibition zones were recorded, and the percentage of mycelium growth inhibition was calculated by using the following formula: inhibition (%) = (C − T/C) × 100, where C is the growth in the control and T is the growth in the treatment. Plates without bacterial inoculated fungal disks were used as controls, and each treatment was performed in triplicate [12].

### 2.6. Biocontrol Insights into KKD1 In Vitro Responses to Infection in Wheat

The biocontrol insights of KKD1 in vitro responses to *F. graminearum* infection in wheat were also detected. The wheat seeds were surface sterilized, soaked in the bacterial suspension (1 × 10^7^ cfu/mL), sown in basins, and then cultured in a greenhouse (25 °C, photoperiod 16 h/8 h). Leaves of 7-day-old wheat plants were inoculated with the *F. graminearum* mycelium in the middle of the leaf in vitro, following which the wheat leaves were incubated in a sterilized culture incubator at 25 °C for a week. Sterile water treatments were used as the control, and each experiment was performed with three replicates. The reaction result was monitored by microscopic observation of the treated leaves [13].

### 2.7. Extraction of Lipopeptides

Strain KKD1 was cultured with shaking at 180 rpm for 48 h at 37 °C in Landy medium. The culture was then centrifuged at 10,000× *g* at 4°C for 10 min to obtain 100 mL cell-free supernatants. The supernatant was kept overnight at 4 °C after adjusting its pH to 2.0. The precipitates were collected by centrifugation at 10,000× *g* at 4 °C for 10 min, dissolved in methanol, and the pH was readjusted to 7.0. The methanolic extracts were passed through a 0.2 µm filter [14].

### 2.8. MALDI-TOF-MS

The matrix-assisted laser desorption/ionization time-of-flight mass spectrometry (MALDI-TOF-MS) analysis used the ribosomal protein spectra provided in the proprietary database, and the results were performed in linear positive mode. The molecular masses of the lipopeptide compounds were analyzed by MALDI-TOF-MS with a Bruker Daltonik Reflex MALDI-TOF instrument with a 337 nm nitrogen laser for desorption and ionization. A-Cyano-4-hydroxycinnamic acid served as the matrix [15].

### 2.9. HPLC Analysis

High-Performance Liquid Chromatography (HPLC) was used for the analysis of the lipopeptides. The lipopeptide mixture was filtered through a 0.2 µm membrane filter. A microbore 1200 system (Agilent Technologies, Santa Clara, CA, USA) with an Agilent Eclipse XDB-C18 5 µm column (Macherey-Nagel, Duren, Germany) was used to separate the isoforms with a flow rate of 1 mL/min and an injection volume of 20 µL [16].

### 2.10. Environmental Stress

Salt stress resistance was examined by inoculating 1 mL of liquid culture (cultured for 12 h at 37 °C) into 100 mL LB broth containing increasing NaCl concentrations (3–15%) for 24–96 h. The OD_600_ of the liquid culture was examined at 12 h, 24 h, 48 h, 72 h, and 96 h. LB broth was used as the control. Each experiment was performed with three replicates [3].

Alkaline stress resistance was examined by inoculating 1 mL liquid culture (cultured for 12 h at 37 °C) into 100 mL LB adjusted to pH 7.0–11.0. The OD_600_ of the bacterial suspension was examined at different time intervals, including 12, 24, 48, 72, and 96 h. LB broth was used as the control, and each experiment was performed with three replicates [6].

### 2.11. Quantitative Real-Time PCR

The samples were collected at various stress levels to observe the expression of tolerance-related genes. RNA was extracted by the Bacterial RNA extraction kit (OMEGA Bio-tek, Inc. Norcross, GA, USA), cDNA was synthesized by The OneScript^®^ cDNA Synthesis Kit (Applied Biological Materials, Inc. Zhenjiang, China). The primers were designed by the Primer Quest tool of IDT (Appendix A). The SYBR Green qPCR master mix (TakaraBio, Changping District, Beijing, China) was used for PCR reactions, the expression of genes was studied in a Real-time Thermocycler (QuantStudio-6 Thermo Fisher Scientific, San Jose, CA, USA) by the program of 95 °C for 30 s, 40 cycles of 95 °C for 5 s, and 34 s at 60 °C. The relative expression levels of all samples were calculated and analyzed based on the 2^−∆∆CT^ method [16].

### 2.12. Primers, Sequence Alignment, RNA Analysis, and Statistical Analysis

All the PCR primers relevant to this study were designed by SnapGene v4.1.9. The sequences of the *Bacillus* species were downloaded from the NCBI database (with a similarity criterion of >95%). The multiple sequence alignment was performed by ClustalW. The concentration and purity of the RNA were determined by measuring the absorbance at 260/280 nm (NanoDrop 1000, Thermo Scientific, Wilmington, DE, USA) and then statistically analyzed by one-way ANOVA and Duncan’s test in SPSS Statistics version v17.0 (SPSS Inc., Chicago, IL, USA) and Excel 2019 (Microsoft Corp., Redmond, WA, USA). Significance was assessed at *p* ≤ 0.05. The graphs were edited by GraphPad Prism 8.0.1, Adobe Photoshop 2021, and Adobe Illustrator 2021.

## 3. Results and Discussion

### 3.1. Isolation and Identification of the Stress-Tolerant Bacillus spp. from the QTP

Utilizing microbes from the QTP as biopesticide is a reasonable approach for the development of an environmentally friendly biological control approach to sustainable agriculture under harsh environmental conditions [3]. A total of 25 strains were isolated from the rhizospheric soil sampled in different extreme habitats (deserts, saline-alkali soil, and farmlands) of the QTP (Appendix A). The average altitude of these sampling sites was 3025 m. While 16S rRNA sequences are not sufficient to distinguish between closely related species of the *B. subtilis* species complex, *gyrB* sequences yield sufficient resolution of closely related species. Therefore, the partial *gyrB* sequences (Appendix A) were used for the taxonomic identification. In the phylogenetic tree (Figure 1), the strains clustered into four different species: *Bacillus atrophaeus* (19 strains), *Bacillus velezensis* (two strains), *Bacillus thuringiensis* (two strains), *Bacillus subtilis* (one strain) and *Bacillus halotolerans* (one strain). The number of *B. atrophaeus* strains selected from the extreme environment of the QTP was remarkable. Due to its promising stress resistance, *B. halotolerans* KKD1 was selected for further analysis. KKD1 was isolated from the rhizosphere of *Androsace mbellate* in Hoh Xil Natural Nature reserve (34°19′ N and 89°25′ E, 4680 m above sea level), which is located in the northeastern part of the QTP. The sampling site is characterized by alkaline soil, high salinity, a low annual temperature (−4° to −9 °C), and high UV radiation.

### 3.2. The Genome of B. halotolerans KKD1

To further explore the genes features of KKD1, we assessed the whole-genome analysis. Genome sequencing and mining supported the phenotypic features observed in *B. halotolerans* KKD1 at the molecular level. Whole-genome sequence analysis of *B. halotolerans* KKD1 was accomplished using the Illumina sequencing platform (see Methods). The KKD1 genome (Figure 2) consisted of a 4,248,134-bp circular chromosome with an average GC content of 43.57%. KKD1 encoded 4465 predicted protein-coding sequences (CDSs), 85 tRNA genes, 30 rRNAs, and 28 tandem repeat sequences. In total, the 3307 CDS sequences comprised 25 functional COGs (Appendix A), 42 GO categories (Appendix A), and 30 KEGG metabolic pathways (Appendix A). Twenty-one sRNAs were identified, eight of which were related to stress-resistance and biocontrol traits, and their target mRNA binding sites are described in Appendix A. In total, 441 genes in *B. halotolerans* KKD1 clustered into nine different DNA islands and were probably acquired via horizontal gene transfer. Some of these gene products exhibited striking similarity to sodium: proton antiporter and UV damage repair protein on the amino acid sequence level. In addition, *B. halotolerans* KKD1 harbored six different confirmed CRISPR-Cas systems, which were defined as acting against invading genetic elements.

Comparison of *B. halotolerans* KKD1 with the model strains *B. velezensis* FZB42 and *B. subtilis* 168 revealed that the three genomes had 3122 core genes in common, but 588 genes were unique to *B. halotolerans* KKD1 (Figure 3A). Structural genome analysis with the Mummer software revealed that *B. halotolerans* KKD1 and *B. subtilis* 168 share a conserved genomic structure and homologous gene composition, but some inversion events also occurred in the genome of these two strains during evolution (Figure 3B–D). On the contrary, *B. halotolerans* KKD1 and *B. velezensis* FZB42 showed a less conserved genomic structure and homologous gene composition (Figure 3E,F).

### 3.3. Taxonomical Assignment of Bacillus halotolerans KKD1 by Its Core Genome

The core genome of KKD1 was used for phylogenomic analysis, applying the EDGAR software package [17]. According to the phylogenetic tree based on the whole genomes of the *B. subtilis* species complex, *B. halotolerans* KKD1 is a member of the *B. subtilis* species complex sensu stricto, as shown in the upper part of Figure 4. Two clusters formed by the *B. subtilis*/*B. vallismortis*/*B. tequilensis* group and by *B. mojavensis*/*B. halotolerans* appear to be closely related. By contrast, *B. velezensis* KKLW is a member of the more distantly related operational *B. amyloliquefaciens* group consisting of *B. amyloliquefaciens*, *B. siamensis*, and *B. velezensis* [18]. It should be noted that *B. subtilis*, *B. licheniformis*, and *B. pumilus* were originally defined as being the first members of the *B. subtilis* species complex [19]. Due to the high-resolution data offered by modern phylogenomic trees based on whole genomes, today the *B. subtilis* species complex contains numerous members, and these three species appear to be very remote from each other.

In order to analyze the taxonomical relationships of KKD1 more precisely, the average nucleotide identity based on BLAST (ANIb) was applied. According to the recommended species cut-off, defined as being 96% [19], KKD1 was found to be a representative of *B. halotolerans*, while KKLW belongs to the species *B. velezensis* (Figure 5). This is consistent with our results obtained by *gyrB* sequence analysis described in Section 3.1.

### 3.4. B. halotolerans KKD1 Tolerates Different Environmental Stress Conditions

The *B. halotolerans* strain KKD1 grew well at 13% salinity, while the model strains *B. velezensis* FZB42 and *B. subtilis* 168 were unable to grow under that condition (Figure 6C). Moreover, The *B. halotolerans* strain KKD1 grew well under alkaline conditions (pH 10.0), while *B. subtilis* 168 only grew only weakly, and *B. velezensis* FZB42 did not grow (Figure 6F). This is in contrast to the taxonomically related model strains *B. velezensis* FZB42 [8] and *B. subtilis* 168 [21], which also belong to the *B. subtilis* species complex [19].

### 3.5. Genome Mining for Genes Possibly Involved in Tolerance to Stress Conditions

#### 3.5.1. Signal Transduction Pathways Involved in Salt Tolerance

We speculate that its extreme habitat might endow *B. halotolerans* KKD1 with stronger self-selectivity at the genetic level. Through genome mining, we found that *B. halotolerans* KKD1 harbored multiple pathways for tolerating complex and diverse stress conditions. To further understand the salt stress resistance mechanism of KKD1 at the molecular level, the genomic features were screened using the GO, COG, and KEGG databases, and key genes involved in these different pathways were compared. The expression of genes involved in these pathways was analyzed using RT-qPCR (Figure 7A).

The presence of genes related to sodium transporters suggested that one strategy by which KKD1 endures high salinity is by maintaining cellular homeostasis by transporting sodium. Gene0757 probably encodes sodium/proline symporter, it is the homologue of *opuE* in *B. subtilis* and will be called *opuE*. The expression of *opuE* and *putP* was upregulated 15.91- and 10.46-fold, respectively. The sodium/proline symporters *opuE* and *putP* are members of the solute/sodium symporter (SSS) family. They are essential membrane integrated proteins that couple the flow of Na^+^ ions driven by electrochemical Na^+^ gradients to the transport of proline across biological membranes [22]. Gene2041 probably encodes sodium: proton antiporter, it is the homologue of *nhaC* in *B. mycoides* and will be called *nhaC.* The expression of *yuiF*, *nhaC*, gene0525 and gene2258 was upregulated 11.68-, 8.71-, 4.88- and 2.52-fold, respectively. These four genes are sodium: proton antiporters and sodium: glutamate symporters, which are responsible for maintaining cellular homeostasis by exchanging protons for sodium ion.

The presence of genes involved in various biosynthesis activities suggested that another strategy by which KKD1 endures high salinity is by accumulating compatible solutes. Proline, betaine, and trehalose are the main compounds maintaining osmotic balance in cells [23]. Glycine betaine is one of the most important osmotic-regulating chemicals and it can accumulate rapidly in cells to maintain osmotic balance and protect pivotal enzymes under salt stress. The genes involved in glycine/betaine metabolism, namely, *proV*, *proW*, *proX*, *opuA* and *opuD* were highly expressed (3.89- to 27.12-fold); they are all glycine/betaine ABC transporters. The glycine/betaine ABC transporters are dimerized by binding to ATP and transferring the substrate to the other side of the membrane by changing their conformation [24]. Based on the K00130 and the K00128 pathways in KEGG, *betB* was annotated as betaine-aldehyde dehydrogenases, which can activate glycine, serine, and threonine metabolism; it upregulated 5.89-fold when KKD1 was exposed to 13% salinity. Trehalose is a nonreducing disaccharide synthesized by trehalose synthase, which protects cells and bioactive substances [25]. The expression of trehalose permease IIC protein TreB and trehalose operon TreR2 regulator was also enhanced (6.95- to 22.36-fold). Proline is a compatible osmotic regulator, we found that the expression of these genes involved in the proline pathway, namely *proS* and gene0381 (PRODH 2) was highly upregulated (7.60- to 34.15-fold). Gene0381 encodes proline dehydrogenase, it is the homologue of PRODH 2 in *B. marisflavi*, and will be called PRODH 2. Proline dehydrogenase (PRODH) also known as proline oxidase (POX), the first step in the two-step oxidation of proline in bacteria. It plays the biological role in cell homeostasis and adapts through energetic, physiological and adaptive processes in eukaryotes [26]. Additionally, the proline-tRNA ligase ProRS and the proline dehydrogenase act together to decrease intracellular salt stress pressure [22].

Based on this result, we speculated that the histidine signal transduction pathways act as the sensory network and the two-component systems act as basic stimulus–response coupling mechanisms to sense environmental stress [6]. KKD1 might eliminate Na^+^ and absorb various solute to keep the intracellular NaCl concentration lower than the external environment and thus survive under high salt conditions [27]. Further research is necessary to elucidate their action mechanism in *B. halotolerans* KKD1 in more detail.

#### 3.5.2. Alkaline Resistance

Based on the results obtained by the KEGG functional annotation of *B. halotolerans* KKD1, we hypothesized that the bacterium response to alkaline pressure was mainly due to export, catabolism, environmental adaptation, and signal molecular interaction.

Resistance and pH homeostasis proteins (MnhA-G) were found in the K05565 signaling pathway. The expression of these genes was upregulated (2.54- to 3.14-fold) under alkaline stress (Figure 7B). They maintain the normal growth of cells by regulating intracellular osmosis. On the test results, the expression of *yuiF* was upregulated significantly 5.62-fold. The *yuiF* is a Na^+^/H^+^ antiporter, which is involved in the K03315 and K07084 signaling pathways in KEGG. The Na^+^/H^+^ antiport activity is primarily aimed at extruding Na^+^ to the external space, and the Na^+^/H^+^ antiporter YuiF imports extracellular H^+^ to exchange intracellular Na^+^, thus acidifying the cytoplasm [28]. We also found that *urtD* upregulated 2.56-fold in *B. halotolerans* KKD1 under alkaline stress. It is reported that the ATP-binding cassette (ABC) transporter *urtD* maintains normal homeostasis by transporting amino acids and ions [29]. Na^+^/H^+^ antiporters and ABC transporters showed the same response when *B. halotolerans* KKD1 is under salt and alkaline stress [30]. The expression of *tuaG* and *tuaH* was upregulated 2.61- and 2.97-fold, respectively. The glycosyl transferase TuaG and the teichuronic acid biosynthesis glycosyl transferase TuaH are required for polymerizing teichuronic acid. The production of teichuronic acid enables cells to repel OH^−^ and adsorb more Na^+^ and H^+^ [31]. All of these genes maintained the intracellular pH at a lower level than the extracellular pH by concentrating the dissolved substances in the cytoplasm.

In addition, through the comparative genomics analysis, we noticed that KKD1 harbors specific genes involved in stress resistance, thus providing a new perspective on the genomic features of extremophilic *Bacillus* species. *B. halotolerans* KKD1 contains four genes related to proton transportation, cytoplasm acidification, and signal transduction. Gene2977 probably encodes the sodium ABC transporter ATP-binding protein; it is the homologue of *bcrA* in *B. subtilis*, and we renamed it as the *bcrA*. The solute: sodium symporter family transporter (gene2258), and sodium: dicarboxylate symporter (gene0525) and the two-component sensor histidine kinase VanS, do not occur in the model strains FZB42 and 168. We speculate that these proteins are involved in sensing and response of environmental signals, such as the transport of extracellular H^+^ to exchange intracellular Na^+^, and the acidification of the cytoplasm, so further regulating the balance of osmotic pressure inside and outside of the cell. The function of these genes requires further experimental verification and we expect to find novel genetic features that elucidate the mechanisms of stress resistance occurring in the extremophilic representatives of the genus *Bacilllus*.

### 3.6. Biocontrol of Plant Pathogens In Vitro

In order to investigate the biocontrol potential of KKD1 in vitro, we proved its antifungal activity by choosing *Fusarium graminearum* [12]. The fungal pathogen *F. graminearum* has a broad host range, causing scab in major cereal crops, such as wheat, barley, and maize, and is responsible for substantial economic losses every year. A strong suppressive effect was detected on agar plates containing *F. graminearum* when it was co-cultured with KKD1. The average size of the inhibition zone was ≥11 mm (Figure 8B). In addition, the lipopeptide extracts of KKD1 were also efficient in inhibiting *F. graminearum* (Figure 8C). Subsequently, *B. halotolerans* KKD1 was assessed for its ability to reduce the lesion diameter of wheat leaves. Compared with the control, the growth of the lesion diameter was very slow on the wheat leaves sprayed with a bacterial suspension of KKD1. The biocontrol effect of KKD1 visibly inhibits *F. graminearum* infection in wheat. Our results demonstrated that *B. halotolerans* KKD1 was able to partially overcome the pathogenic effects and had the potential to inhibit the development of plant disease (Figure 8A).

### 3.7. Genome Mining for Gene Clusters Devoted to the Synthesis of Antimicrobial Secondary Metabolites

Our genome mining results revealed that KKD1 possesses numerous genes involved in the production of bioactive compounds [32]. AntiSMASH version 6.0 [33] was used to predict the gene clusters in *B. halotolerans* KKD1 possibly involved in the control of plant disease (Figure 8). In total, KKD1 harbors 10 different secondary metabolite gene clusters, three of which were not detected in the model strains FZB42 and 168. Six of the biosynthetic gene clusters (BGCs) in KKD1 are listed in the MIBiG data bank [34] and are common in the representatives of the *B. subtilis* species complex. Producing compounds encoded by these gene clusters is considered the main route by which *B. halotolerans* KKD1 controls plant disease [35].

We assume that the antifungal activity of KKD1 relies mainly upon the non-ribosomal-synthesized lipopeptides fengycin and surfactin. Both are cyclopeptides linked to a ß-fatty acid chain [36]. The lipopeptides surfactin, fengycin/plipastatin, and bacillibactin are synthesized non-ribosomally and have been previously detected in *B. subtilis* [37]. Surfactin is a lipoheptapeptide with antimicrobial, antiviral, and anti-mycoplasma activity. The 25 kb *srf* operon in *B. halotolerans* KKD1 was similarly organized as in *B. subtilis* 168. Fengycin is a cyclic lipodecapeptide with antifungal action. Surfactin contributes to inducing systemic resistance against plant pathogens [38], and fengycin can act directly against pathogenic fungi by destabilizing their membrane integrity, although its role in stimulating the ISR response in plants cannot be excluded [8].

The siderophore bacillibactin can absorb and accumulate Fe ions under iron limitation and has an important competitive role with other microorganisms [12]. Functional *dhb* and *ent* gene clusters involved in the synthesis of bacillibactin were also detected in *B. halotolerans* KKD1. The dipeptide bacilysin (l-alanine-[2,3-epoxycyclohexano-4]-l-alanine) is efficient in the suppression of plant pathogenic bacteria. Bacilysin was encoded by the *bacABCDEFG* gene cluster in *B. halotolerans* KKD1.

Polyketide synthases (PKSs) are involved in the non-ribosomal synthesis of polyketide antibiotics [8]. The pks gene cluster present in *B. halotolerans* KKD1 encoded bacillaene, an open-chain enamine acid with extended polyene structure that is known for its antagonistic effect against bacteria [39]. In contrast to the lipopeptides and polyketides described above, lanthipeptides are ribosomally synthesized and post-translationally modified peptides containing lanthionine or labionin structures [40]. The sactipeptide subtilosin A, an example of an unusual lantibiotic, has a macrocyclic structure with three thioether bonds. The gene cluster for subtilosin A synthesis was detected in the genome of KKD1, and we also detected that peptidase M16 was part of the subtilosin A biosynthetic gene cluster in KKD1. It inhibits Gram-positive bacteria and is expressed under stress conditions [3]. In addition, KKD1 harbored three gene clusters, whose products have not yet been identified. One biosynthesis gene cluster in the genome of *B. halotolerans* KKD1 contains a protein kinase/lanthionine synthetase C (LanC) family protein. Class III lanthipeptides with a molecular weight of 1738 Da are encoded by five gene copies located downstream from the *lanC* gene. Type III polyketide synthase (T3PKS) is also common in a multitude of *Bacillus* species and might contribute to its antagonistic activity [3]. A chalcone and stilbene synthase protein is encoded in the KKD1 genome. Terpene occurs in nature with varied structural compounds and has been reported to possess antimicrobial activity [8]. The products of two terpene biosynthesis gene clusters presented in *B. halotolerans* KKD1 may contribute to its bacteriostatic and bactericidal effects [41].

The gene clusters involved in synthesis of secondary metabolites of strain *B. halotolerans* KKD1 and model strain *B. velezensis* FZB42 were compared, and huge differences were found. For comparison, we have also included the results of our comparative analysis performed with *B. velezensis* KKLW, which isolated from the same harsh environment (QTP) as KKD1. According to the corresponding NCBI genome neighbor report, the genome sequence of KKLW (CP054714.1) was found to be nearly identical to the model strain FZB42 (CP000560.2). During our analysis, the gene clusters involved in synthesis of secondary metabolites are identical in KKLW and FZB42, despite the fact that they were isolated from very different habitats, suggesting that the constitution of secondary metabolites among KKD1 and FZB42 was caused by the species differences, not the environmental factors.

It is known that the representatives of the *B. amyloliquefaciens* operational group [18] possess a higher diversity of secondary metabolites than other members of the *B. subtilis* species complex. In fact, the plant-associated *B. velezensis* FZB42 has devoted more than 10% of its genomic capacity to the synthesis of secondary metabolites with antimicrobial action [8]. Our AntiSMASH-supported analysis of the *B. velezensis* FZB42 and KKLW genome corroborated the occurrence of additional gene clusters not present in *B. halotolerans* KKD1 (Figure 9). In addition to bacillaene, the genomes of FZB42 and KKLW harbor two additional giant gene clusters devoted to the non-ribosomal synthesis of the type I trans-AT polyketides macrolactin and difficidin. Macrolactin and difficidin might be considered unique phylogenetic markers for the species *B. velezensis* [42]. Difficidin in particular is a powerful antibacterial metabolite [14]. Another gene cluster devoted to the non-ribosomal synthesis of the cyclic lipopeptide bacillomycin D, not occurring in KKD1, was detected in the genomes of FZB42 and KKLW. Bacillomycin D is known as the main antifungal acting metabolite in the plant-associated model strain FZB42 [8]. Gene clusters for the ribosomal synthesized bacteriocins (RiPPs) plantazolicin and amylocyclicin were detected in the genome of KKLW and FZB42, but not in the KKD1 genome [43].

A comparison of all gene clusters involved in the non-ribosomal and ribosomal synthesis of secondary metabolites with antimicrobial action detected by antiSMASH analysis of the KKD1, FZB42 and KKLW genomes is presented in Appendix A.

### 3.8. Detection of Surfactin and Fengycin in the Supernatant of KKD1

The lipopeptides present in the culture supernatant of several *Bacillus* species are known to inhibit plant pathogens. The *B. halotolerans* KKD1 strain developed antifungal activity against *F. graminearum*, as described in Section 3.6. The expression of surfactin and fengycin in KKD1 was experimentally validated by HPLC and MALDI-TOF-MS. We extracted the crude lipopeptides from strain KKD1 and analyzed the main components by HPLC and MALDI-TOF-MS. Two lipopeptide peaks were detected by HPLC (Figure 9). We purified the compounds obtained from these two peaks and analyzed their components by MALDI-TOF-MS. The molecular weight of the lipopeptides was estimated to be in the range of 900–2000 kDa (Figure 10A,B). The molecular masses of the ions at *m*/*z* 1030.59 corresponded to C_13_ surfactin A [M + Na]^+^, at *m*/*z* 1074.60 to C_16_ surfactin A [M + Na]^+^, at *m*/*z* 1030.59 to C_13_ surfactin A[M + Na]^+^, at *m*/*z* 1486.71 to C_16_ fengycin A [M + Na]^+^, and at *m*/*z* 1501.69 to C_14_ fengycin B [M + K]^+^.

## 4. Conclusions

Using microorganisms from the QTP as biofertilizer is a suitable technique for developing of environmentally friendly biologicals that may be used in agriculture in harsh conditions. The extremophilic *B. halotolerans* KKD1 isolated from Hoh Xil possesses a strong biological potential to plant pathogens, and can propagate under high salinity and alkalinity stresses. *B. halotolerans* KKD1 contains four unique genes involved in different mechanisms of stress tolerance which not occurring in the model strains FZB42 and 168: sodium symporter family transporter (gene2258), and the sodium: dicarboxylate symporter (gene0525). Moreover, KKD1 also exhibits promising biocontrol activity against plant pathogens due to its production of surfactin, fengycin, and other secondary metabolites such as different bacteriocins. In conclusion, the result revealed that *B. halotolerans* KKD1 could be developed as a promising biological agent under harsh environmental conditions.

## Figures and Tables

**Figure 1 biology-10-01030-f001:**
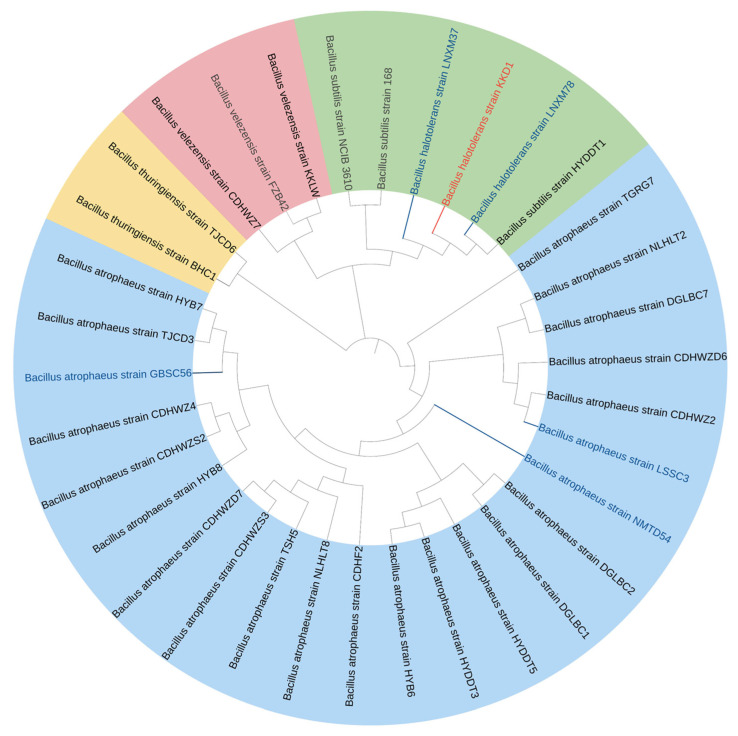
Phylogenetic tree based on the alignment of partial *gyrB* nucleotide sequences. The strains were clustered into four different groups: *B. atrophaeus*, *B. velezensis*, *B. halotolerans*, and *B. thuringiensis*. *B. halotolerans* KKD1 is labelled by red letters, and the model strains are labelled by grey letters. Strains previously isolated from the Qinghai–Tibetan Plateau are labelled by blue letters [3].

**Figure 2 biology-10-01030-f002:**
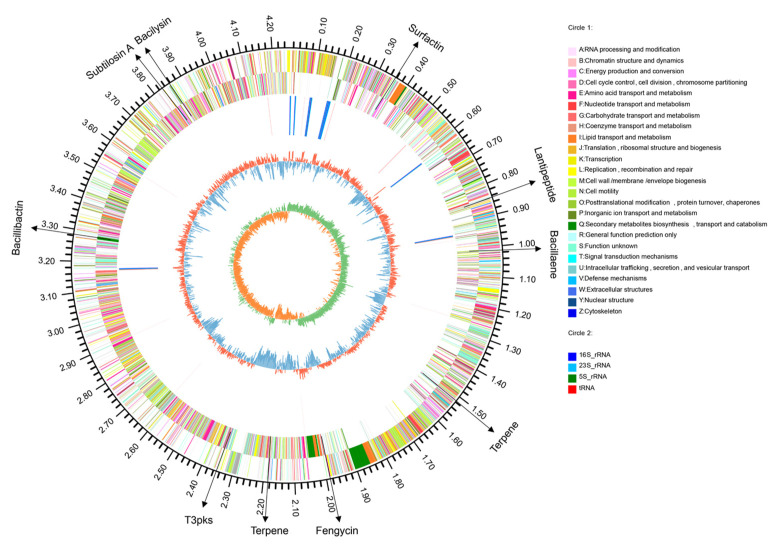
The whole-genome map of *B. halotolerans* KKD1. The genome map consists of seven different circles. From the inner circle to the outer circle: (1) GC skew. (2) G + C content. (3) The locations of the secondary metabolite clusters. (4) Reverse COG function classification. (5) Forward COG function classification. (6) Reverse CDS. (7) Forward CDS.

**Figure 3 biology-10-01030-f003:**
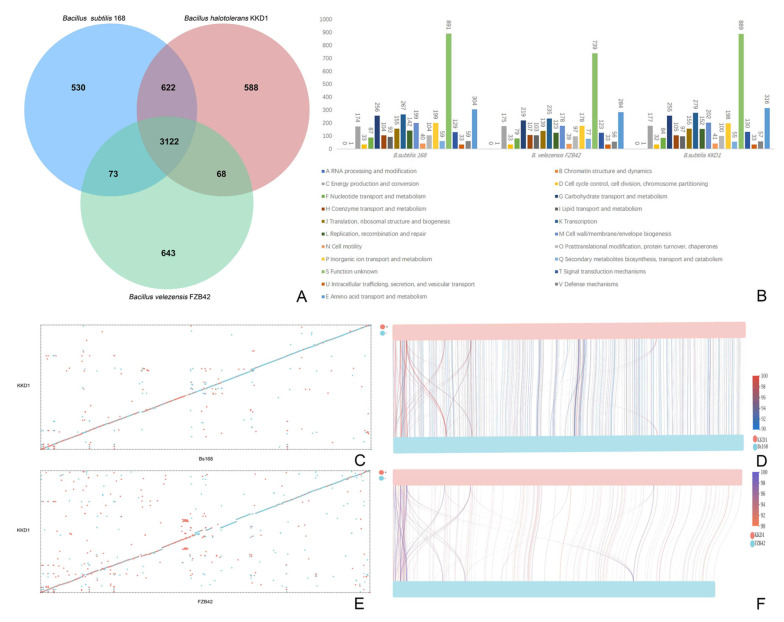
Comparative genomic analysis of KKD1 with model *Bacillus* strains. (**A**) Each strain is represented by an oval. The number of orthologous genes shared by all strains (*B. halotolerans* KKD1, *B. velezensis* FZB42 and *B. subtilis* 168) is in the center. Numbers in nonoverlapping portions of each oval show the number of CDSs unique to each strain. (**B**) Genome analysis of the three strains based on the COG database. (**C**) A dot plot of the collinearity analysis between *B. halotolerans* KKD1 and *B. subtilis* 168. The abscissa represents *B. subtilis* 168, and the ordinate represents *B. halotolerans* KKD1. (**D**) The line graph of collinearity analysis between *B. halotolerans* KKD1 and *B. subtilis* 168. (**E**) A dot plot of collinearity analysis between *B. halotolerans* KKD1 and *B. velezensis* FZB42. The abscissa represents *B. velezensis* FZB42, and the ordinate represents *B. halotolerans* KKD1. (**F**) The line graph of collinearity analysis between *B. halotolerans* KKD1 and *B. velezensis* FZB42. Note: The blue dots in (**C**,**E**) represent the forward comparison of two strains, the composition and direction of the sequences are the same, and the red dots represent the reverse comparison of two samples. The sequence composition is the same, and the direction is opposite. Different colors in (**D**,**F**) indicate different regions in the samples, and the color and width of the color band indicate the alignment length and the region homology.

**Figure 4 biology-10-01030-f004:**
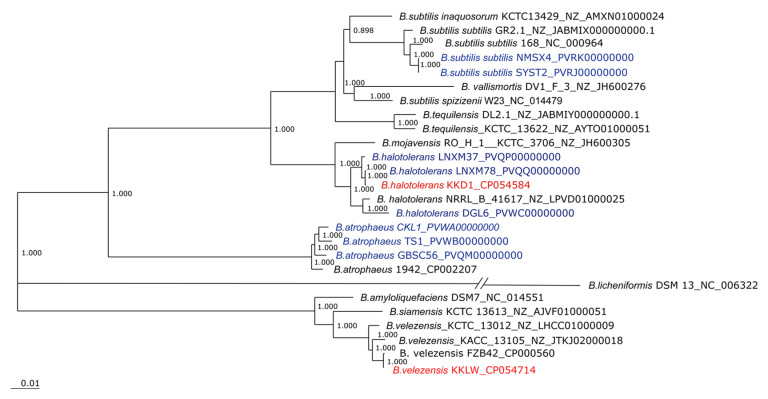
Phylogenetic tree. Phylogenetic tree for 26 genomes constructed from a core of 1743 genes per genome (45,318 in total). The core has 519,198 AA-residues/bp per genome and 13,499,148 in total. *B. halotolerans* KKD1 is labelled by red letters. Strains previously isolated from the Qinghai–Tibetan plateau are labelled by blue letters. *B. licheniformis* DSM13 was used as an outgroup. FastTree software was implemented with the EDGAR package to generate an approximately-maximum-likelihood phylogenetic tree. The values at the branches of the FastTree trees are not bootstrapped values but are local support values computed by FastTree using the Shimodaira-Hasegawa test (FastTree, http://www.microbesonline.org/fasttree/; accessed on 04 October 2021).

**Figure 5 biology-10-01030-f005:**
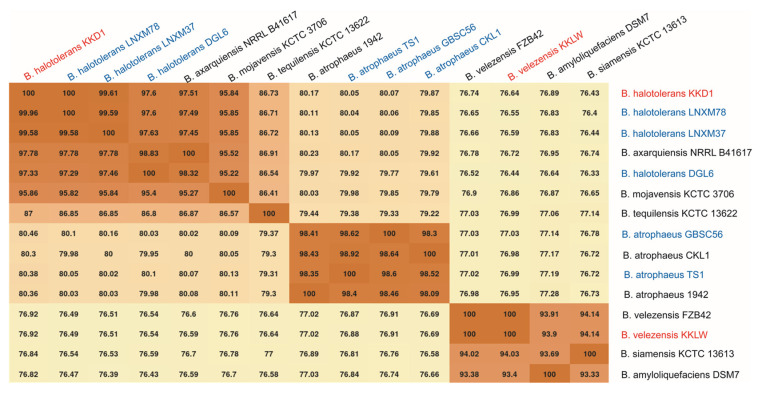
Heatmap of the ANIb matrix calculated for the genomes of *B. halotolerans* KKD1. Heatmap of the ANIb matrix calculated for the genomes of *B. halotolerans* KKD1 and closely related species. The method was based on a BLASTN comparison of the genome sequences as described by Goris et al. [20]. *B. halotolerans* KKD1 is labelled in red letters. Strains isolated from the Qinghai–Tibetan plateau are labelled in blue letters.

**Figure 6 biology-10-01030-f006:**
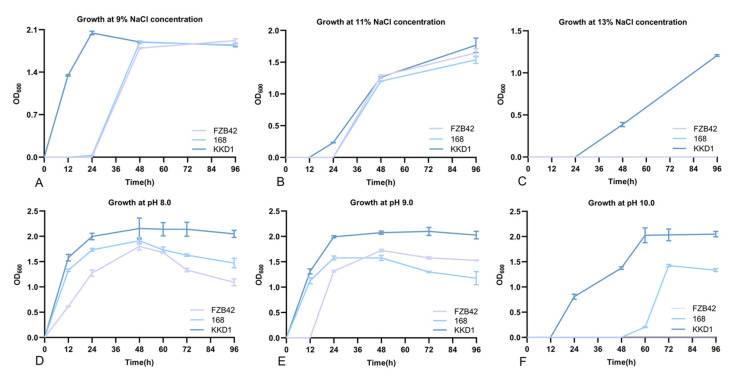
Growth curve of *B. halotolerans* KKD1 under different stress conditions. Growth curve under salt stress (9%, 11%, 13%) and growth curve under alkaline stress (pH 8.0, pH 9.0, pH 10.0). The error bars represent the mean standard deviation of each treatment repeated three times with three replicates (**A**–**F**).

**Figure 7 biology-10-01030-f007:**
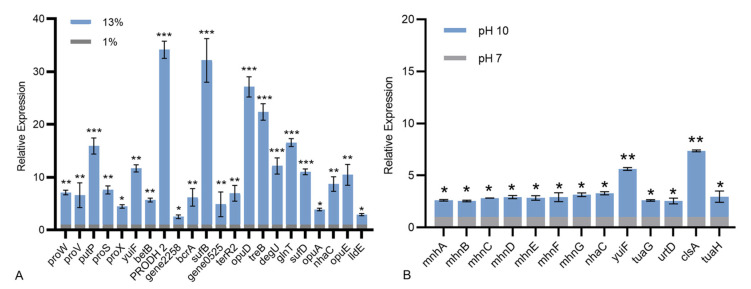
Relative expression levels of different genes involved in stress resistance. (**A**) The relative expression levels of different genes in *B. halotolerans* KKD1 grown at 1% and 13% salinity. (**B**) The relative expression levels of different genes in *B. halotolerans* KKD1 grown at pH 7.0 and pH 10.0. Data are the average thickness ± the standard deviation (SD) of three independent experiments. *, indicates a significant difference (*p < 0.05*), **, indicates an extremely significant difference (*p < 0.05*), ***, indicates an extremely significant difference (*p < 0.01*) compared to that of control group.

**Figure 8 biology-10-01030-f008:**
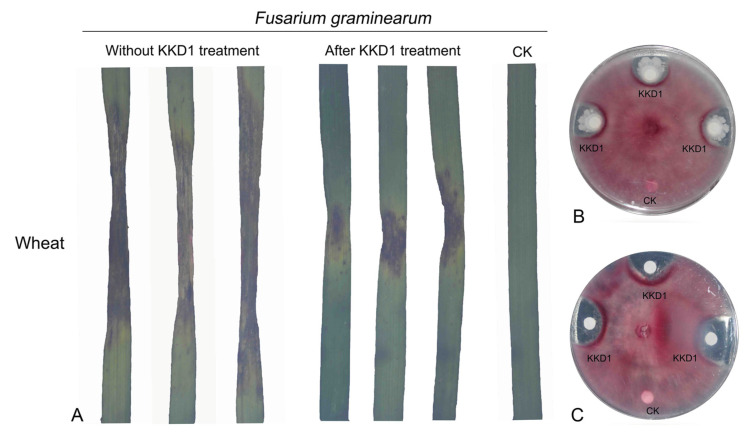
Antagonistic action of *B. halotolerans* KKD1 versus *F. graminearum*. (**A**) In vitro response of wheat leaves after infection with *F. graminearum*. Representative lesion lengths in wheat leaves following inoculation with control *F. graminearum* and *F. graminearum* after cultivation with strain KKD1. (**B**) Inhibition zones of *F. graminearum* caused by the KKD1 colonies. (**C**) Inhibition zones of *F. graminearum* caused by the lipopeptide extracts of KKD1.

**Figure 9 biology-10-01030-f009:**
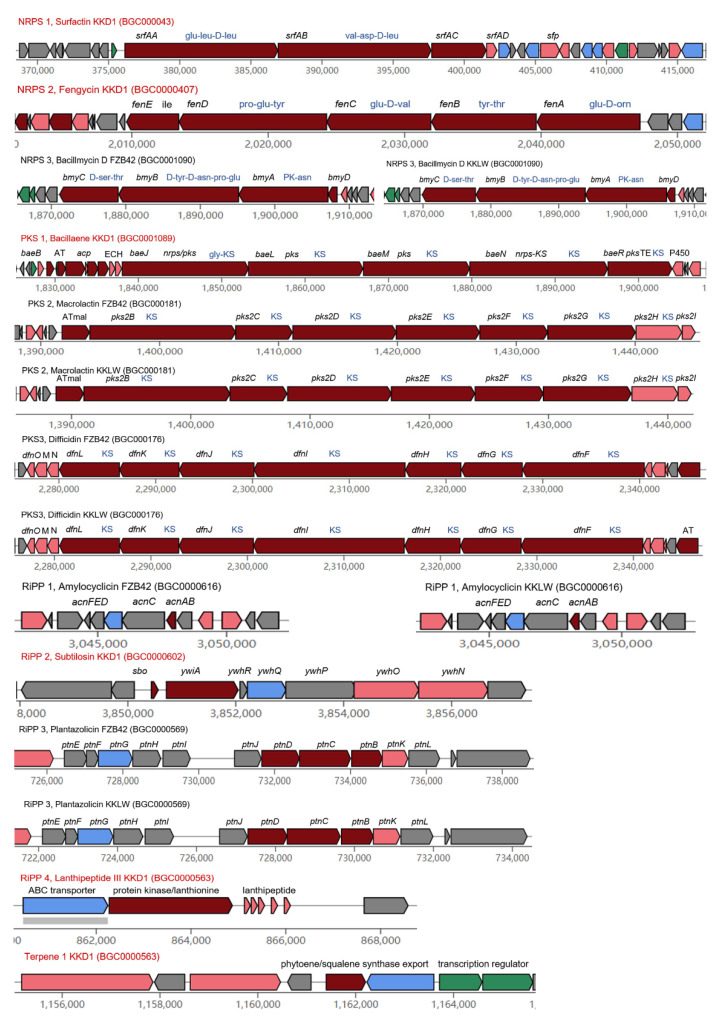
Thebiosynthetic gene clusters involved in the antagonistic activity of KKD1, FZB42 and KKLW. Detection of the gene clusters devoted to the synthesis of secondary metabolites was accomplished with antiSMASH 6.0. The genome of *B. halotolerans* KKD1 contains the gene clusters for the non-ribosomal synthesis of surfactin (NRPS), fengycin/plipastatin (NRPS), bacillibactin (NRPS), and bacilysin (other). Trans-AT-polyketide-synthases (PKS) type I accomplish the non-ribosomal synthesis of bacillaene. Chalcone/stilbene synthesis is accomplished by type III polyketide synthase (T3PKS). Additional gene clusters involved in the non-ribosomal synthesis of secondary metabolites are present in *B. velezensis* FZB42 and KKLW: bacillomycin D (NRPS), macrolactin (PKS), and difficidin (PKS). Ribosomal synthesis of peptides with antagonistic action against competing microorganisms occurs in KKD1, such as subtilosin, and uncharacterized lanthipeptide II. FZB42 and KKLW are characterized by ribosomal synthesis of RiPPs amylocyclicin and plantazolicin, which were previously described in *B. velezensis* FZB42.

**Figure 10 biology-10-01030-f010:**
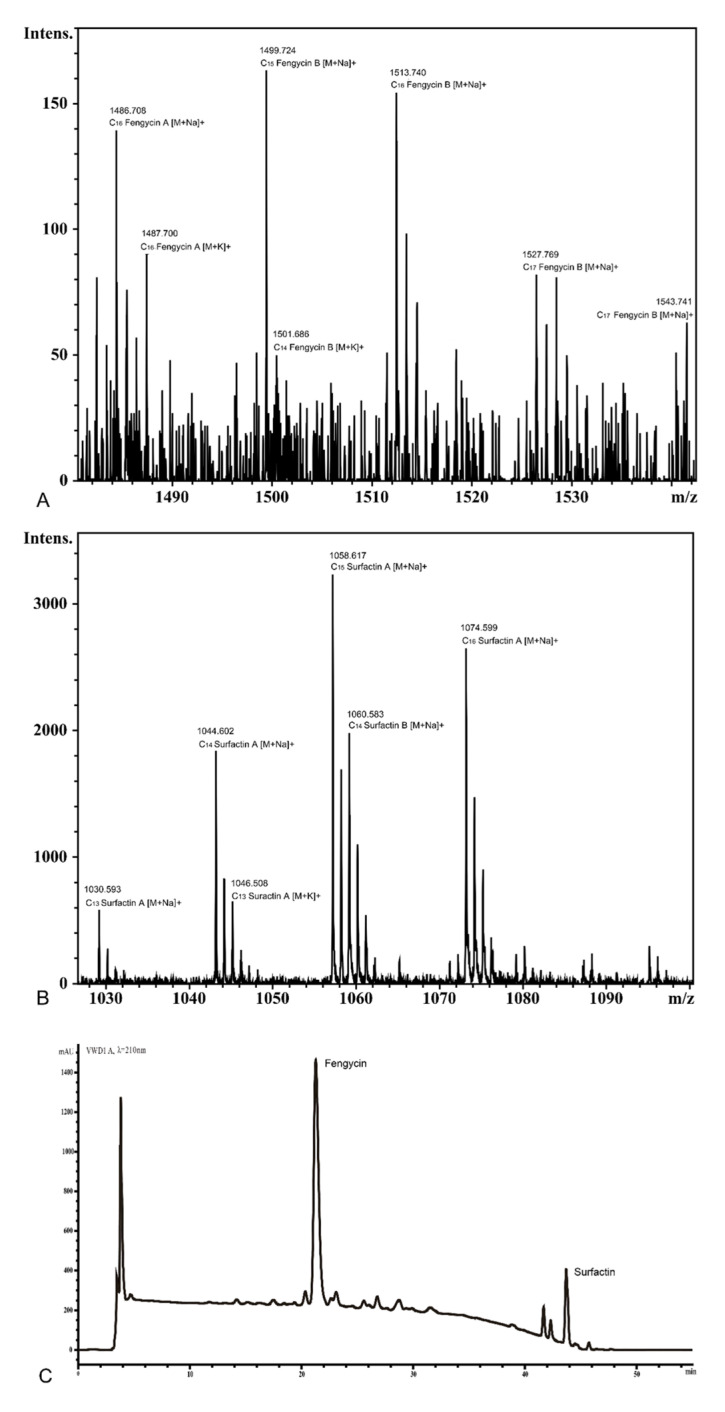
MALDI-TOF-MS and HPLC analysis of lipopeptide in *B. halotolerans* KKD1. (**A**) MALDI-TOF-MS analysis of surfactin in *B. halotolerans* KKD1. (**B**) MALDI-TOF-MS analysis of fengycin in *B. halotolerans* KKD1. (**C**) HPLC analysis of lipopeptide in *B. halotolerans* KKD1.

## Data Availability

Not applicable.

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
