# Peer review of "Genomic Features and Molecular Function of a Novel Stress-Tolerant Bacillus halotolerans Strain Isolated from an Extreme Environment"

_biology, 2021, doi:10.3390/biology10101030_

Round 1

Reviewer 1 Report

In the analyzed rhizospheric soil samples, the authors identified 19 Bacillus atrophaeus strains, 2 Bacillus velezensis strains, 2 Bacillus halotolerans strains, and 2 Bacillus thuringiensis strains. Because of its promising stress resistance, B. halotolerans KKD1 was selected for further analysis.

First of all, I was disappointed by the poor quality of figures in the main text of the manuscript. However, authors identified strain candidates that can be used as biocontrol strains in the future and can be of interest for the community. The experimental approach and results obtained are interesting but are devaluated by the low quality of the presented figures. For this manuscript to be suitable for publication the quality of the figures must be improved significantly.

In several parts (e.g. lines 240-242,258-259, 384-397) of the work strain B. velezensis KKLW is mentioned and discussed. It contains additional gene clusters coding secondary metabolites not present in B. halotolerans KKD1 (lines 384-397) so why authors did not pick this strain for detailed analysis?

In lines 221 and 222 I miss explanation of COG, GO, KEGG abbreviations.

In Fig. 5 I would suggest to include not just average values but also the deviations in growth curves. Authors mentioned these experiments were repeated 3 times, so it should not be problem.

 In chapters 3.5.1. and 3.5.2 several genes annotated only with numbers are discussed. Are there some homologues of these gene products among other bacteria? If yes, for the better reading I suggest to annotate these genes by names of the corresponding homologues.

Namely:

gene0757 mentioned just as sodium/proline symporter

genes 2041,0525, and 2258 which should code sodium ABC transporter ATP-binding proteins, sodium: proton antiporter, sodium: dicarboxylate symporter, and solute: sodium symporter family transporter, respectively

gene2271 identified as betaine-aldehyde dehydrogenase,

gene0757, and gene0381 of proline pathway, if possible specify in more detail

gene2977, which possibly encodes the sodium ABC transporter ATP-binding protein

In chapter 3.6 I miss positive control in the experiment and from the poor quality of the provided figure, it is not possible to clearly see the reduction of the damage caused by F. graminearum on wheat leaves.

The chapters 3.7 and 3.8 I could not evaluate due to bad quality of the figures attached.

The discussion part is not very sound and is just a brief summary of presented conclusions and would more fit to Conclusion part.

Author Response

Dear Editors and Reviewers:

Thank you very much for your helpful suggestions and valuable input in our research manuscript entitled “Genomic features and molecular function of a novel stress-tolerant Bacillus halotolerans strain isolated from an extreme environment(ID: biology-1353019). We also very much appreciate the suggestions made by reviewers. According to the suggestions, we have revised whole manuscript and did improvement during revision throughout the manuscript. We also incorporated most of the suggestions during our revision. Accordingly, a point-by-point response is provided and the revisions are highlighted in the main text with red color and main correction in the manuscript and responds to reviewer’s comments separately. We hope the revised version of manuscript is now more suitable and fit to standard of authoritative journal for publication.

Point to Point response to Reviewer 1:

Response: Thank you very much for your helpful suggestions and valuable input in our research manuscript. We also very much appreciate the suggestions made by reviewers. According to the suggestions, we have revised, whole manuscript and did improvement during revision throughout the manuscript. We also incorporated most of the suggestions during our revision. Accordingly, a point-by-point response is provided and revisions are highlighted in the main text with red color.

Point. 1

First of all, I was disappointed by the poor quality of figures in the main text of the manuscript. For this manuscript to be suitable for publication the quality of the figures must be improved significantly.

Response: Thank you very much for your useful comments. Unfortunately, due to the compression connected with the PDF format, the quality of the figures became unacceptable. So here, we provide the original manuscript for your review.

Point. 2

In several parts (e.g. lines 240-242,258-259, 384-397) of the work strain B. velezensis KKLW is mentioned and discussed. It contains additional gene clusters coding secondary metabolites not present in B. halotolerans KKD1 (lines 384-397) so why authors did not pick this strain for detailed analysis?

Response:  In our previous study, we also did the detailed analysis of strain KKLW, which showed significant ability to inhibit various plant pathogen, but cannot withstand high salinity and alkaline conditions. Therefore, we only compared here the gene clusters involved in coding secondary metabolites between B. velezensis KKLW and B. halotolerans KKD1. Furthermore, we have studied the differences of metabolites produced by different strains. As result, we corroborated earlier findings that macrolactin and difficidin should be considered as unique phylogenetic markers for the B. velezensis species. Overall, KKD1 did not only show significant antagonistic activity, and was more stress-resistant than KKLW and other related strains. So, we have selected KKD1 for further detailed studies.

Point. 3

In lines 221 and 222 I miss explanation of COG, GO, KEGG abbreviations.

Response: Thanks, the explanation of COG, GO and KEGG abbreviations is mentioned in Material and Method part (line 138-140). COG (Cluster of Orthologous Groups of Proteins, http://www.ncbi.nlm.nih.gov/COG), KEGG (Kyoto Encyclopedia of Genes and Genomes, https://www.kegg.jp/), GO (Gene Ontology, http://geneontology.org/).

Point. 4

In chapters 3.5.1. and 3.5.2 several genes annotated only with numbers are discussed. Are there some homologues of these gene products among other bacteria? If yes, for the better reading I suggest to annotate these genes by names of the corresponding homologues.

Response: Thank you for your valuable comment, yes, these genes are homolog to other known genes, so we renamed the genes by their ‘RecName' which is recommended by the Swissprot database: gene0757 encodes sodium/proline symporter, and  is the homologue of opuE in B. subtilis, we renamed it as the opuE; gene0381 encodes proline dehydrogenase, the homologue of PRODH 2 in B. marisflavi, we renamed it as the PRODH 2; gene2041 encodes sodium: proton antiporter, the homologue of nhaC in B. mycoides,we renamed it as nhaC; gene2271 encodes betaine-aldehyde dehydrogenase, the homologue of dhaS in B. subtilis, we renamed it as dhaS; gene2977 encodes the sodium ABC transporter ATP-binding protein, the homologue of bcrA in B. subtilis, we renamed it as bcrA. The detailed function of gene0525 (sodium: proton antiporter) and gene2258 (solute: sodium symporter family transporter) is unknown but was annotated according to their conserved domain. Therefore, no RecName is available until now. We plan to do a detailed study of these two genes to determine their functions and then rename them according to their functions.

Point. 5

Gene0757, and gene0381 of proline pathway, if possible specify in more detail.

Response: Thanks, as reviewer suggested, we have annotated these genes more detail in the manuscript in line 305-307 and 326-330.

The gene0757 encodes sodium/proline symporter, is a member of the SSS family of sodium/solute transporters. Sodium/proline symporters are essential membrane integrated proteins that couple the flow of Na+ ions driven by electrochemical Na+ gradients to the transport of proline across biological membranes (Hoffmann T et.al. 2012). It protects the cells by balancing the osmotic pressure inside and outside, and helps to improve salt tolerance of the bacteria enabling them to grow under high salt conditions.

The gene0381 encodes proline dehydrogenase, the homologue of PRODH 2 in Bacillus spp. Proline dehydrogenase (PRODH) also known as proline oxidase (POX), the first step in the two-step oxidation of proline in bacteria. It is involved in cell homeostasis, and in different adaptive processes in eukaryotes (Servet C et.al. 2012).

Point. 6

In chapter 3.6 I miss positive control in the experiment and from the poor quality of the provided figure, it is not possible to clearly see the reduction of the damage caused by F. graminearum on wheat leaves.

Response: Thanks, due to the compression of PDF format, the figures are blurred. here, we provide the original manuscript for your review (see Fig. 7). The lesion lengths of wheat leaves treated with KKD1, and F. graminearum, were reduced in comparison to the control only treated with F. graminearum. Shanshan Xie et.al. (2018) used the same method to prove the antagonistic ability of Bacillus strain against Xanthomonas oryzae of rice leaves. Therefore, we referred to that method to test the prevention effects of KKD1 on wheat leaves against F. graminearum.

Point. 7

The chapters 3.7 and 3.8 I could not evaluate due to bad quality of the figures attached.

Response: Due to the compression of PDF format, the figures are blurred and the quality deteriorates. We provided now the original manuscript for your review (see Fig. 8 and 9).

Point. 8

The discussion part is not very sound and is just a brief summary of presented conclusions and would more fit to Conclusion part.

Response: Thanks, according to reviewer suggestion we have modified the discussion part, please see the discussion part in the manuscript.

Reviewer 2 Report

  • The authors have raised an important question about the genomic features and molecular function of stress-tolerant Bacillus halotolerans and have followed a rational and logical approach to determine these factors.

Major comment:

  • The biggest challenge for me was to follow the figures. The figures are low resolution and blur and not at all publication quality. Why have authors not used high resolution images. This has made it very difficult for me to review this manuscript and appreciate the science behind it.

Minor comment:

  • Growth assay: In Fig 5, it is not clear to me why do authors claim that model strains are unable to grow in harsh conditions. Most of them easily grow till 1.5. Am I missing something here? Again, it is hard to differentiate between different strains of Bacillus because of the image quality.

Author Response

Dear Editors and Reviewers:

Thank you very much for your helpful suggestions and valuable input in our research manuscript entitled “Genomic features and molecular function of a novel stress-tolerant Bacillus halotolerans strain isolated from an extreme environment(ID: biology-1353019). We also very much appreciate the suggestions made by reviewers. According to the suggestions, we have revised whole manuscript and did improvement during revision throughout the manuscript. We also incorporated most of the suggestions during our revision. Accordingly, a point-by-point response is provided and the revisions are highlighted in the main text with red color and main correction in the manuscript and responds to reviewer’s comments separately. We hope the revised version of manuscript is now more suitable and fit to standard of authoritative journal for publication.

Point to Point response to Reviewer 2:

Response: Thank you very much for your helpful suggestions and valuable input in our research manuscript. We also very much appreciate the suggestions made by reviewers. According to the suggestions, we have revised whole manuscript and did major improvement during revision throughout the manuscript. We also incorporated most of the suggestions during our revision. Accordingly, a point-by-point response is provided and revisions are highlighted in the main text with red color.

Point. 1

The biggest challenge for me was to follow the figures. The figures are low resolution and blur and not at all publication quality. Why have authors not used high resolution images. This has made it very difficult for me to review this manuscript and appreciate the science behind it.

Response: Thank you very much for your useful comments. In related to the quality of our figures is relatively high, but due to the compression of PDF format, the figures are blurred and the quality deteriorates. So here, we provide the original manuscript for your review.

Point. 2

Growth assay: In Fig 5, it is not clear to me why do authors claim that model strains are unable to grow in harsh conditions. Most of them easily grow till 1.5. Am I missing something here? Again, it is hard to differentiate between different strains of Bacillus because of the image quality.

Response: Thanks, due to the compression of PDF format, the figures are blurred. So here, we provide the original manuscript for your review.

The B. halotolerans strain KKD1 grew well at 13% salinity, while the model strains B. velezensis FZB42 and B. subtilis 168 were unable to grow under that condition (Fig. 5C). Moreover, The B. halotolerans strain KKD1 grew under alkaline conditions (pH 10.0), while B. subtilis 168 only grew weakly, and B. velezensis FZB42 did not grow (Fig. 5F).

Round 2

Reviewer 1 Report

I’d like to thank the authors for the answers to my questions. However, the manuscript still needs extensive re-writing or data addition, as well as language and style correction prior publication. I'm sorry but I have to insist that it should be submitted only after extensive editing as its present quality is not in accordance with the standards requested by the journal. The results are interesting but the presentation is not very sound.

My main reservations/suggestions:

  1. The manuscript contains some vague claims and need to be formulated more precisely. E.g. Line 247 “Some of these genes exhibited striking similarity to sodium: proton antiporter and UV damage repair protein.” - Is this similarity on nucleotide or amino acid sequence level?
  2. The authors did not reflect to my question/suggestion concerning the Fig. 5. Authors should provide graphs which include standard deviations not only average values of the measurements.
  3. I also suggest to include adapted explanation provided by the authors about the genes gene0757, gene0381, gene2041, gene2271, gene2977, gene0525 and gene2258 mentioned in version 1 of the manuscript to corresponding positions in the main text. E.g gene0757 probably encodes sodium/proline symporter, it is the homologue of opuE in B. subtilis and will be called opuE. Etc.
  4. The paragraph 6. Biocontrol of plant pathogens in vitro needs some introduction explaining the motives of the authors, which led to this experiment.
  5. I suggest to exclude the details about velezensis KKLW antimicrobial secondary metabolites (ASM) from the manuscript or include data or citation, if available, on its resistance to high pH and alkalic conditions. For the readers it is still not clear from the text why it was not picked as the strain to be studied in more detail. If the authors want to compare ASM of the B. halotolerans KKD1 to some strain, I suggest to compare it rather to model strain B. velezensis FZB42, which was used as a control in previous experiments.
  6. Most importantly, the Discussion part needs most extensive re-writing. It contains several copied sentences from results. Some sentences are even duplicated within Discussion itself, which is unacceptable. The authors should stress out why their finding are important, not just repeat the results.

Author Response

Thank you very much for your helpful suggestions and valuable input in our research manuscript. According to the suggestions, we have revised the whole manuscript and did major improvement during the revision. Accordingly, a point-by-point response is provided and revisions are highlighted in the manuscript with “Track Changes” function.

Point. 1

The manuscript contains some vague claims and need to be formulated more precisely. E.g. Line 247 “Some of these genes exhibited striking similarity to sodium: proton antiporter and UV damage repair protein.” - Is this similarity on nucleotide or amino acid sequence level?

Response: Thanks, to be more precise we have changed the sentence as following: “Some of the gene products exhibited striking similarity to sodium: proton antiporter and UV damage repair protein on amino acid sequence level (see line 257-259).

Point. 2

The authors did not reflect to my question/suggestion concerning the Fig. 5. Authors should provide graphs which include standard deviations not only average values of the measurements.

Response: Thanks, we have made correction according to the Reviewer’s comments (See Figure 5).

Point. 3

I also suggest to include adapted explanation provided by the authors about the genes gene0757, gene0381, gene2041, gene2271, gene2977, gene0525 and gene2258 mentioned in version 1 of the manuscript to corresponding positions in the main text. E.g gene0757 probably encodes sodium/proline symporter, it is the homologue of opuE in B. subtilis and will be called opuE. Etc.

Response: Thanks, we have changed and provided the explanation of these genes in the manuscript (See line 347-348, line 352-354, line347-346, line 413-416).

Point. 4

The paragraph 6. Biocontrol of plant pathogens in vitro needs some introduction explaining the motives of the authors, which led to this experiment.

Response: Thanks, well raised comments. We have modified the introduction of this experiment to explain our motives in this part. In order to investigate the biocontrol potential of KKD1 in vitro we proved its antifungal activity by choosing Fusarium graminearum (See line 430-431).

Point. 5

I suggest to exclude the details about B. velezensis KKLW antimicrobial secondary metabolites (ASM) from the manuscript or include data or citation, if available, on its resistance to high pH and alkalic conditions. For the readers it is still not clear from the text why it was not picked as the strain to be studied in more detail. If the authors want to compare ASM of the B. halotolerans KKD1 to some strain, I suggest to compare it rather to model strain B. velezensis FZB42, which was used as a control in previous experiments.

Response: Thanks, we have changed and provided the comparison of the ASM in KKD1 with FZB42. The reason why we have included KKLW for comparison with KKD1 was that the strain has been isolated from the same environment as KKD1. Meanwhile, the sequences of KKLW and FZB42 are nearly identical. During our analysis, the gene clusters involved in synthesis of secondary metabolites are identical in KKLW and FZB42, despite that they were isolated from very different habitats, suggesting that the constitution of secondary metabolites among KKD1 and FZB42 was caused by the species differences, not the environmental factors.

Point. 6

Most importantly, the Discussion part needs most extensive re-writing. It contains several copied sentences from results. Some sentences are even duplicated within Discussion itself, which is unacceptable. The authors should stress out why their finding are important, not just repeat the results.

Response: Thanks, according to the reviewer’s suggestion we have rewritten the discussion part and compile with results (Results and discussion). During the revision we did extensive revision and improve English language of our manuscript.

Reviewer 2 Report

Dear authors,

Thank you for providing the answers.

Author Response

Dear reviewer,

Thank you very much for your helpful suggestions in our research manuscript.

Round 3

Reviewer 1 Report

I'd like to thank the authors for the improvement of their manuscript and for answering my questions. I hope that in the final version of the manuscript the  inserted figures are going to be in printing quality because again in the version 3 the figs are unreadable.